# Brood care in a 100-million-year-old scale insect

Bo Wang[1,2]*, Fangyuan Xia[3], Torsten Wappler[2], Ewa Simon[4], Haichun Zhang[1], Edmund A Jarzembowski[1,5], Jacek Szwedo[6]

[1]State Key Laboratory of Palaeobiology and Stratigraphy, Nanjing Institute of Geology and Palaeontology, Chinese Academy of Sciences, Nanjing, China; [2]Steinmann Institute, University of Bonn, Bonn, Germany; [3]Nanjiao Bieshu 394, Shanghai, China; [4]Department of Zoology, University of Silesia, Katowice, Poland; [5]Department of Earth Sciences, Natural History Museum, London, United Kingdom; [6]Department of Invertebrate Zoology and Parasitology, University of Gdańsk, Gdańsk, Poland

**Abstract** Behavior of extinct organisms can be inferred only indirectly, but occasionally rare fossils document particular behaviors directly. Brood care, a remarkable behavior promoting the survival of the next generation, has evolved independently numerous times among animals including insects. However, fossil evidence of such a complex behavior is exceptionally scarce. Here, we report an ensign scale insect (Hemiptera: Ortheziidae), *Wathondara kotejai* gen. et sp. nov., from mid-Cretaceous Burmese amber, which preserves eggs within a wax ovisac, and several freshly hatched nymphs. The new fossil is the only Mesozoic record of an adult female scale insect. More importantly, our finding represents the earliest unequivocal direct evidence of brood care in the insect fossil record and demonstrates a remarkably conserved egg-brooding reproductive strategy within scale insects in stasis for nearly 100 million years.

## Introduction

Brood care is an altruistic trait that evolved to enhance the fitness of offspring at a cost to the parents and represents a breakthrough in the adaptation of organisms to their environment (*Tallamy, 1984*; *Clutton-Brock, 1991*; *Gilbert and Manica, 2010*). Fossil evidence of such an ephemeral behavior is extremely rare, reported mainly in dinosaurs (*Meng et al., 2004*; *Varricchio et al., 2008*), ostracods (*Siveter et al., 2007*, *2014*), arachnids (*Engel and Grimaldi, 2014*), but rarely in insects. Until now only two putative examples in Mesozoic insects have been described based on fossils (*Peñalver et al., 2012*; *Cai et al., 2014*). Although phylogenetic analyses suggest some ancient insects evolved brood care (e.g., *Korb et al., 2012*), only fossils provide unequivocal direct evidence. In this study, we report on an exceptionally preserved insect from mid-Cretaceous Burmese amber, which represents the earliest unequivocal direct evidence of brood care in the insect fossil record and sheds new light on the early evolution of such behavior.

## Results

### Systematic paleontology

Order Hemiptera Linnaeus, 1758.
 Family Ortheziidae Amyot and Serville, 1843.
 *Wathondara kotejai* gen. et sp. nov. Simon, Szwedo and Xia.

### Etymology

The generic name refers to Wathondara—goddess of earth in Buddhist mythology from Southeast Asia. Gender: feminine. The species is named after the late Polish entomologist Jan Koteja in recognition of his significant contribution to the study of both extant and fossil scale insects.

*For correspondence:
savantwang@gmail.com

Competing interests: The authors declare that no competing interests exist.

**eLife digest** Many animals care for and protect their offspring to increase their survival and fitness. Insects care for their young using a range of strategies: some dig underground chambers for their young, whilst others carry their brood around on their own bodies. However, it was unclear when these strategies first evolved in insects.

Now Wang et al. report that they have discovered the earliest fossil evidence of an insect caring for its young, in the form of a female insect preserved with her brood in a specimen of ancient amber. The amber comes from northern Myanmar, where amber deposits are around 95–105 million years old. The fossilised insect is an adult female scale insect with a cluster of around 60 eggs on her abdomen. Six young scale insect nymphs are also preserved in the same piece of amber. Wang et al. named this newly discovered species *Wathondara kotejai*, after an earth goddess in South-East Asian Buddhist mythology and the late Polish entomologist Jan Koteja.

Most scale insect fossils found to date have been males. Fossilised adult females are scarcer, most likely because female scale insects are wingless and less mobile and therefore less prone to accidental burial. The fossil reported by Wang et al. is therefore a rare find, and it is also sufficiently well preserved to reveal that the female's eggs are contained within a wax-coated egg sac. Today there are many species of scale insects, most of which are parasites of plants and many are economically important pests of trees and shrubs. In living relatives of *W. kotejai*, females use a similar wax coating to protect themselves and their offspring: young nymphs hatch inside the egg sac and remain there for a few days before emerging into the outside world.

This new fossil provides a unique insight into the anatomy and life cycle of a long-extinct insect; it also demonstrates that brood care in insects is an ancient trait that dates back to at least around 100 million years ago at the height of the age of the dinosaurs.

## Holotype

BA14011. The amber piece preserves an adult female with eggs, six first-instar nymphs, and a weevil. It is polished in the form of a flattened ellipsoid cabochon, clear and transparent, with diameter about 11 mm, height about 5 mm, and weight about 0.8 g.

## Locality and age

Specimen is from Kachin Province in northern Myanmar. Burmese amber has been dated biostratigraphically from late Albian to Cenomanian (about 105 to 95 million years old), based on an ammonite and palynology (*Cruickshank and Ko, 2003*; *Ross et al., 2010*). The U-Pb dating of zircons from the volcaniclastic matrix of the amber gave an age of 98.8 ± 0.6 million year (*Shi et al., 2012*).

## Diagnosis (based on adult female)

Body elongate oval, dorsoventrally flattened (seems to be natural condition). Antennae 8-segmented; first segment straight, elongate, thicker than others, trapezoid in shape; second segment cylindrical distinctly longer than others; antennal segments III–VIII with numerous setae of hair-like and fleshy types, some of them almost as long as apical setae on segment VIII. Apical segment cylindrical with long and stout apical seta and additional shorter subapical seta situated on subapical projection. Legs slender; trochanter fused with femur; tibia and tarsus fused, with numerous spine-like setae. Tarsal claw without denticles; claw digitules hair-like, thin, and short.

## Description

Amber specimen preserves adult female with about 60 elliptical eggs (0.3 mm long, 0.2 mm wide) in wax ovisac, and six first-instar nymphs near adult (*Figure 1*). Adult body elongate oval, 6 mm long, 2 mm wide (with ovisac). Antenna about 1.2 mm long, inserted ventrally at frontal margin, with eight segments (*Figure 2C*); first, the widest, trapezoidal; second, the longest, cylindrical; segments III to VII, club-like; segment VIII cylindrical, with subapical projection; length of antennal segments (in mm) I—0.162; II—0.350; III—0.130; IV—0.130; V—0.145; VI—0.115; VII—0.125; VIII—0.220. Segments I

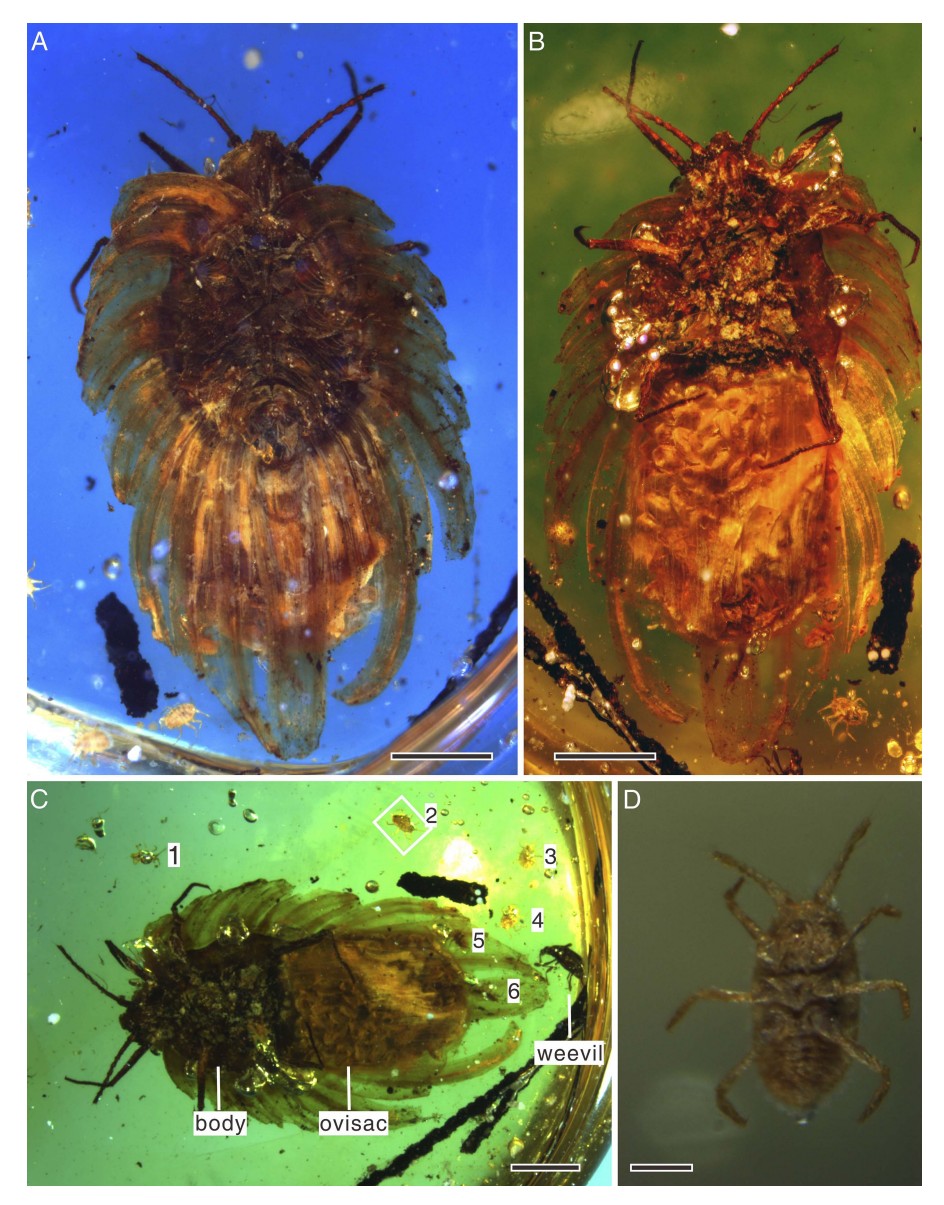

**Figure 1**. *Wathondara kotejai* gen. et sp. nov. Simon, Szwedo and Xia from mid-Cretaceous Burmese amber. (**A**) Habitus in dorsal view, stacked image with a blue filter. (**B**) Habitus in ventral view, stacked image with a green filter. (**C**) Habitus in ventral view, stacked image with a green filter. Note the weevil under the adult. The numbers 1–6 represent six first-instar nymphs. (**D**) Enlargement of a nymph in (**C**). Scale bars of (**A**, **B** and **C**) represent 1 mm; scale bar of (**D**) represents 0.1 mm.

and II covered with scarce hair-like setae, segments III to VII with subapical fleshy setae on external margins and hair-like setae; some setae almost as long as apical setae of VIII segment; segment VIII with subapical seta on projection (0.075 mm long) and apical seta (0.097 mm long). Eyes not easily observable, placed on short stalks. Labium apparently 2-segmented. Legs well-developed; tarsal claw small, slightly bent, without denticle. Anal ring visible on dorsum. Spiracles, wax glands, and most body setae not visible. Wax secretion of ortheziid type, with nine pairs of marginal lobes, two triangular frontal lobes, three elongate triangular median lobes, nine submedian pairs, and posterior lobes (*Figure 3*). Wax covering made partly translucent due to preservation in amber, completely covering dorsum (*Figure 2A*). Ovisac well-developed, 3 mm long, 2.1 mm wide (*Figure 2D*). Six

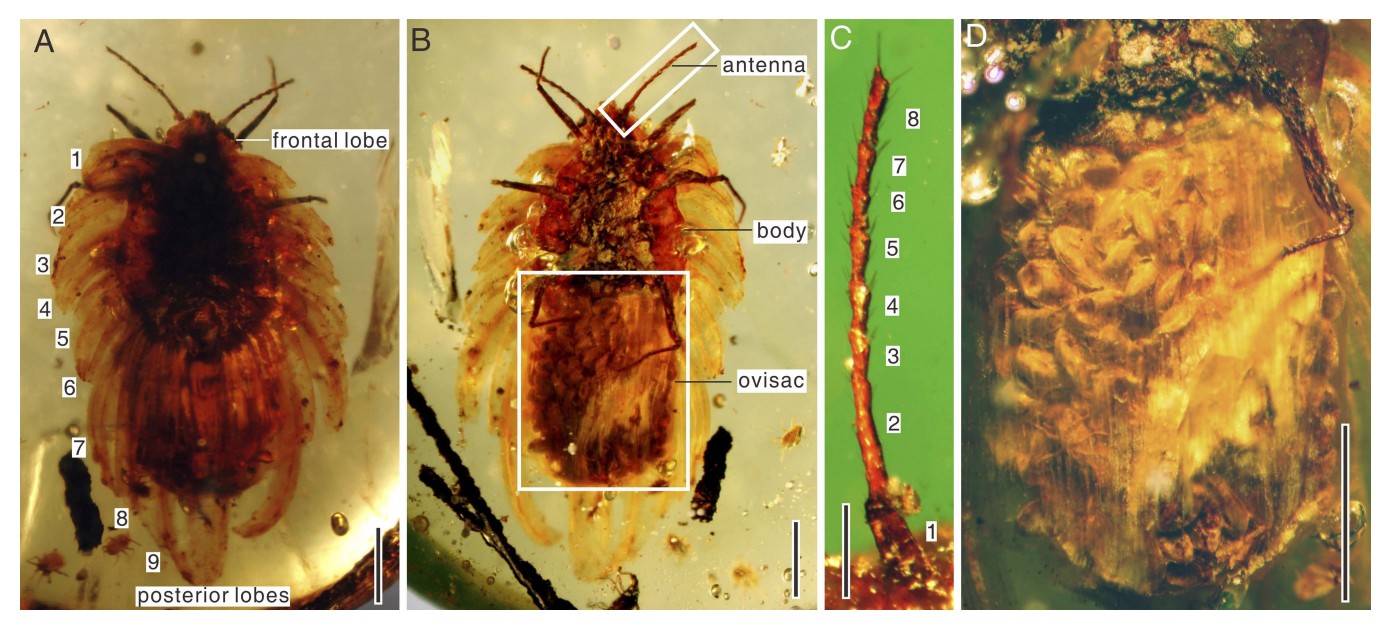

**Figure 2**. *Wathondara kotejai* gen. et sp. nov. Simon, Szwedo and Xia from mid-Cretaceous Burmese amber. (**A**) Habitus in dorsal view. The numbers 1–9 indicate nine marginal wax lobes. (**B**) Habitus in ventral view. (**C**) Enlargement of the antenna in (**B**). (**D**) Enlargement of the ovisac in (**B**). Scale bars of (**A**, **B**, and **D**) represent 1 mm; scale bar of (**C**) represents 0.25 mm.

associated first-instar nymphs are of similar size, 0.3 mm long, 0.2 mm wide, with only 6-segmented antennae (*Figure 1D*).

## Discussion

Scale insects (Coccoidea), with about 7800 species, are highly diverse, and most of them are obligatory plant parasites often of economic importance (*Ben-Dov et al., 2014*). They exhibit many unusual features of morphology, reproduction, and life history and are thus considered as some of the most evolutionarily fascinating organisms amongst insects (*Wappler and Ben-Dov, 2008*; *Hodgson and Hardy, 2013*). The female life cycle involves two or three actively feeding instars prior to the adult stage, and adult females are wingless, resembling the immature stages (*Gullan and Cook, 2007*). In contrast, adult males are delicate, ephemeral insects with simplified wing venation (*Hodgson and Hardy, 2013*). Scale insects separated from their sister-group, the aphids, at least by the Middle Permian based on the earliest occurrence of Aphidomorpha (*Szwedo et al., 2015*), with the fossil record probably extending back to the Middle Triassic (trace fossils in *Figure 4*). However, their fossil record is over-dominated by males entrapped in fossil resins, and fossil adult females are very scarce—probably because they are commonly sedentary or sessile on host plants (*Koteja, 2000*). To our knowledge, the new fossil is the only Mesozoic record of an adult female, the next oldest being from the late Eocene Baltic amber (*Koteja and Żak-Ogaza, 1988*).

*Wathondara kotejai* is unambiguously referable to Ortheziidae, as evidenced by its general habitus with its body covered with wax plates, ensign-like ovisac, stalked eyes, and well-developed legs (*Kozár, 2004*). Furthermore, *W. kotejai* shares two potential synapomorphies with Recent and Tertiary, crown-group Ortheziidae: differentiated apical and subapical setae on the last antennal segment, and trochanter and femur fused (*Vea and Grimaldi, 2012*). Two Cretaceous genera have been tentatively attributed to Ortheziidae: *Burmorthezia* Vea and Grimaldi in mid-Cretaceous Burmese amber is considered as an extinct sister group to the crown-group Ortheziidae (*Vea and Grimaldi, 2012*), while *Cretorthezia* Koteja and Azar in Early Cretaceous Lebanese amber is probably a stem group of scale insects (*Koteja and Azar, 2008*; *Hodgson and Hardy, 2013*) or an extinct group within Ortheziidae (*Vea and Grimaldi, 2015*). Additionally, a putative female (*Cretorthezia* sp.)

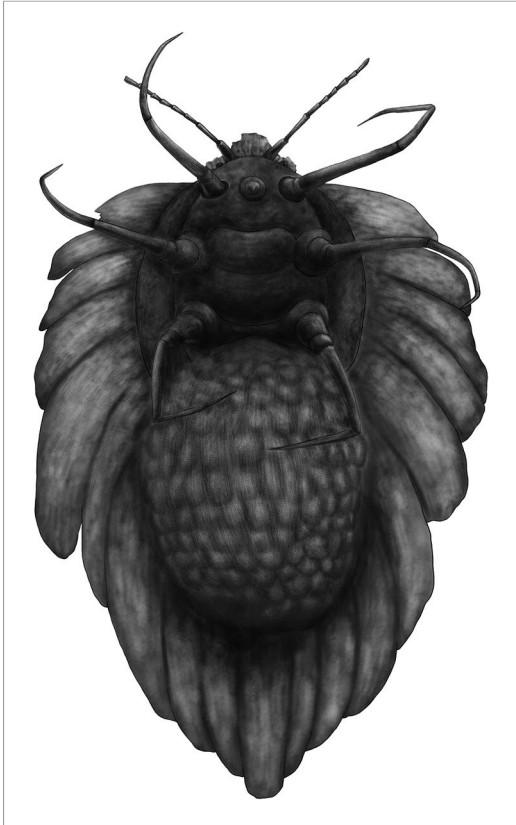

**Figure 3**. Drawing of brooding *Wathondara kotejai* gen. et sp. nov. Simon, Szwedo and Xia in ventral view. The ovisac and wax covering are made nearly transparent by preservation in amber.

from Burmese amber was tentatively identified as an ortheziid, and its systematic position is still uncertain (*Koteja and Azar, 2008*; *Vea and Grimaldi, 2012*). Our new fossil supports the view that crown-group Ortheziidae are present in the mid-Cretaceous.

The seventh, eighth, ninth, and posterior wax lobes of *W. kotejai* are distinctly extended and cover the ovisac dorsally. The thick wax cover not only protects the adult female but also serves to shelter her eggs and first instars. Extant ortheziid females have a band of pores on the ventral side of the abdomen, which secrete a waxy ovisac. The eggs and hatched nymphs are protected within the ovisac (*Figure 3*), as in extant ortheziids and monophlebids (*Vogelsang and Szklarzewicz, 2001*). In extant species, the young nymphs hatch within this ovisac and remain there for a few days until they have acquired a thin covering of wax secretion (visible in our specimens as a slight white pubescence on the fossil nymphs), then crawl out through a split in the wax at the distal end of the ovisac (*Gullan and Kosztarab, 1997*). Extant first instars are mobile and serve as principle agents for dispersion and seeking out suitable feeding sites (*Koteja, 2001*). This egg brooding is widely considered to be a primitive form of brood care (e.g., *Royle, et al., 2012*; *Wong et al., 2013*; *Siveter et al., 2014*). Some Early Cretaceous cockroaches have been reported with an ootheca attached (*Grimaldi and Engel, 2005*, Figure 7.72). However, it is not definitive evidence of egg brooding, because some cockroaches subsequently deposit the ootheca in a suitable crevice. Therefore, *W. kotejai* provides the earliest unequivocal evidence of brood care in insects.

Brood care is considered to have evolved independently in at least seven insect orders (*Wong et al., 2013*). This remarkable behavior takes several forms of which the most common are egg brooding and offspring attendance. Scale insects have evolved a variety of methods to protect their eggs and hatched nymphs from unfavorable abiotic conditions and natural enemies. Some extant species (e.g., Diaspididae, some Pseudococcidae) even possess an ovoviviparous form or pseudoplacental viviparity (*Gullan and Kosztarab, 1997*). In addition to Ortheziidae, ovisacs occur in many other families, for example, Monophlebidae, unrelated Coccidae and many Pseudococcidae, in all of which the secretions of a variety of tubular ducts and disc-pores combine to form the ovisac (*Ben-Dov et al., 2014*). These various types of ovisacs have evolved convergently to protect their offspring from wet and dry conditions, honeydew contamination, and natural enemies (*Gullan and Kosztarab, 1997*). Our study demonstrates that these significant behavioral and morphological adaptations, associated with considerable maternal investment, were already well established by the mid-Cretaceous.

Many extant Ortheziidae females feed on roots and fungal mycelia or mosses and lichens (*Vea and Grimaldi, 2012*) and 'run about' in forest litter with the eggs carried in the ovisac attached to their bodies. This is considered to be the most primitive habit in scale insects (*Gullan and Kosztarab, 1997*; *Koteja, 2001*), and similar brood care behavior also occurs in other early scale insects, for example, Margarodidae and Matsucoccidae (*Koteja, 2001*). Therefore, this behavior probably has an early origin and maybe a synapomorphy for scale insects. Flowering plants and ants are thought to be important drivers for radiation of the most diverse advanced group, the neococcoids (*Grimaldi and Engel, 2005*). However, both factors are absent in the evolutionary history of basal groups of scale

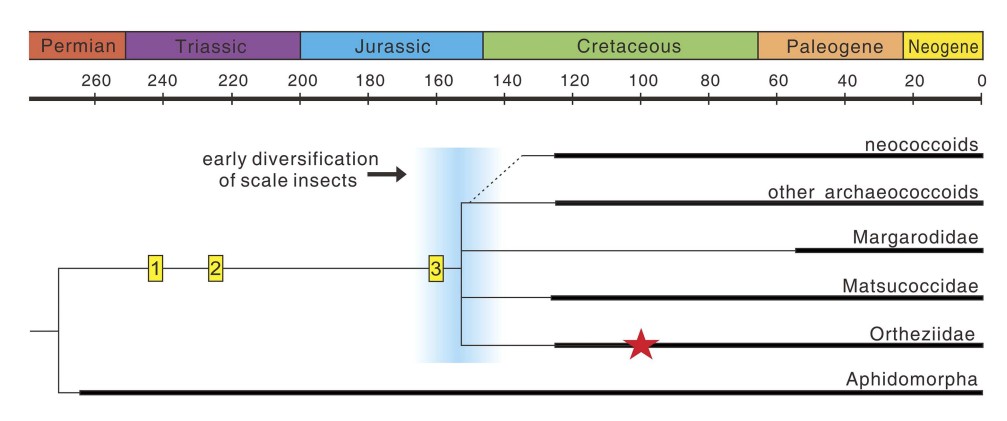

**Figure 4**. The evolution of scale insects. Hypothetical phylogeny based on *Hodgson and Hardy (2013)* and *Vea and Grimaldi (2015)* (extinct families omitted). Matsucoccidae, Ortheziidae, Margarodidae are commonly considered as the most primitive families (*Vea and Grimaldi, 2015*), but their phylogenetic relationships are still unresolved (e.g., *Gullan and Cook, 2007*; *Hodgson and Hardy, 2013*). Thick lines indicate the known extent of the fossil record. (1) Undescribed scale marks on plants from the Middle Triassic Dont Formation of Italy (T Wappler, personal observation, October 2014); (2) scale marks on plants from the Late Triassic Molteno Formation of South Africa (*Labandeira, 2006*); (3) putative, undescribed scale insect from the Late Jurassic (*Grimaldi and Engel, 2005*); Red star represents *Wathondara kotejai* from mid-Cretaceous Burmese amber. An early diversification of scale insects probably occurred during the end of the Jurassic or earliest Cretaceous (blue area), and later radiations are probably closely related to the rise of angiosperms and ants (*Grimaldi and Engel, 2005*).

insects (*Figure 4*). Brood care, greatly promoting the survival of offspring (*Royle et al., 2012*), could therefore have been an important driver for the early radiation of scale insects which occurred during the end of the Jurassic or earliest Cretaceous (*Figure 4*).

Despite a great taxonomic diversity of extant insects with brood care (*Wong et al., 2013*), direct evidence of such behavior has been reported only in Cenozoic ambers (*Peñalver et al., 2012*). The new fossil is unique in providing evidence of ovarian and juvenile developmental stages in a fossil insect. More remarkably, it represents the earliest direct evidence of brood care in insects and highlights the long-term stasis of this behavior in archaeococcoids, extending nearly 100 million years.

## Materials and methods

Burmese amber (amber from northern Myanmar) harbors the most diverse biota in amber from the Cretaceous, and more than 200 families of arthropods have been reported from this deposit. Amber has been recorded from the Shwebo, Thayetmyo, Pakokku, and Pegu districts in Myanmar. However, the only commercial source is the Hukawng Valley in Tanaing Township, Myitkyina District of Kachin State. The amber under study is from an amber mine located near Noije Bum Village, Tanaing Town (*Figure 1* in *Kania et al., 2015*). Four institutes (Nanjing Institute of Geology and Palaeontology, Lingpoge Amber Museum, Huxuan Amber Museum, and Fushun Amber Institute) have collected more than 100,000 amber pieces from this mine, and each piece commonly contains some insects. All these amber pieces were collected from the 'amber layer'. These deposits have been investigated and dated in detail by *Cruickshank and Ko (2003)* and *Shi et al. (2012)*. We tentatively followed the age (98.8 ± 0.6 million years) given by U-Pb dating of zircons from the volcaniclastic matrix of the amber (*Shi et al., 2012*). However, some evidence (e.g., high degree of roundness of amber, bivalve borings on the surface of the amber) suggests that the amber was probably reworked before being deposited in the volcaniclastic matrix.

The type specimen is currently housed in the Nanjing Institute of Geology and Palaeontology (NIGP), Chinese Academy of Sciences and will eventually be deposited in the Lingpoge Amber Museum in Shanghai (specimen available for study by contacting BW or FX). Photographs were taken using a Zeiss SteREO Discovery V20 microscope system. By merging several photographs of one sample, at different focal planes, a single final photograph was created in which the entire sample was

in focus. Blue and green filters were used to improve the contrast between the insect and amber. The figures were prepared with CorelDraw X4 and Adobe Photoshop CS3.

## Nomenclatural acts

The electronic edition of this article conforms to the requirements of the amended International Code of Zoological Nomenclature, and hence the new names contained herein are available under that Code from the electronic edition of this article. This published work and the nomenclatural acts it contains have been registered in ZooBank, the online registration system for the ICZN. The ZooBank LSIDs (Life Science Identifiers) can be resolved and the associated information viewed through any standard web browser by appending the LSID to the prefix 'http://zoobank.org/'. The LSID for this publication is: urn:lsid:zoobank.org:pub: 01114A99-586C-4BAD-9F84-4E3FDBFFD86F. The electronic edition of this work was published in a journal with an ISSN, and has been archived and is available from the following digital repositories: PubMed Central, CLOCKSS, Steinmann Institute at University of Bonn, Natural History Museum (London), University of Gdańsk, and Nanjing Institute of Geology and Palaeontology (CAS).

## Acknowledgements

We are grateful to J Rust, MS Engel, AV Gorochov, and J Chen for helpful discussions, D Yang for the drawing, and two editors and three anonymous reviewers for their constructive reviews.

## Additional information

### Funding

| Funder | Grant reference | Author |
| --- | --- | --- |
| Ministry of Science and Technology of the People's Republic of China | 2012CB821900 | Bo Wang, Haichun Zhang |
| Chinese Academy of Sciences | 2011T2Z04 | Edmund A Jarzembowski |
| Alexander von Humboldt-Stiftung | Research Fellowship | Bo Wang |

The funders had no role in study design, data collection and interpretation, or the decision to submit the work for publication.

### Author contributions

BW, Conception and design, Acquisition of data, Analysis and interpretation of data, Drafting or revising the article, Contributed unpublished essential data or reagents; FX, HZ, Acquisition of data, Analysis and interpretation of data; TW, ES, EAJ, Analysis and interpretation of data, Drafting or revising the article; JS, Acquisition of data, Analysis and interpretation of data, Drafting or revising the article

### Author ORCIDs

Jacek Szwedo, http://orcid.org/0000-0002-2796-9538

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
