## [Decision Letter]

Thank you for sending your work entitled “Brood care in a 100-million-year-old scale insect” for consideration at *eLife*. Your article has been favorably evaluated by Ian Baldwin (Senior editor), a Reviewing editor, and 3 reviewers.

The Reviewing editor has assembled the following comments to help you prepare a revised submission.

The three reviewers appreciated your study. The following main issues were identified and should be addressed in the revision.

1) Whilst it may be true that this record is the first record of actual eggs and young having been found in an insect showing brood care, the importance is slightly over-emphasised. There are earlier records of what are close to the ortheziids (e.g., *Burmacoccus*, from the Early Cretaceous, is pretty clearly an ortheziid, and *Cretorthezia*, also from the Early Cretaceous, is very probably an ortheziid, although these taxa are based on adult males). It is highly probable that these two genera also had adult females with an ovisac, which would surely only be used for brood care. [15] also described an adult female or late nymph that he placed in *Cretorthezia* which looked very similar to extant ortheziids and is therefore very likely to have had an ovisac. Emphasising that *Palaeorthezia* extends crown-group Ortheziidae back only to the mid-Cretaceous therefore seems to overstate their study when there are earlier known probable ortheziid species. This should be explicitly addressed in the manuscript.

2) What is the suggestion about genus/species authorship? Who of the authors of the manuscript will be author of the genus/species? This should be made clear.

3) Figure 4 is based on studies by at least four other sets of workers and some of the studies were not based on any phylogenetic analyses. The tree is therefore hypothetical and this needs to be stated explicitly. Also, *Palaeorthezia* is not apparently included. Surely it should be. But, if it were, it would not be the earliest ortheziid in this figure! You need to include the names of the genera you are referring to in numbers 5 (Early Cretaceous Lebanese amber), 6 (Mid-Cretaceous Burmese amber) and 7 (Eocene Baltic Amber). This would make it much easier for other workers to follow the arguments in this paper.

4) A significant issue that needs to be addressed is the clear delineation of diagnoses for the new genus and species (even if they are the same), and a clear indication of what collection the specimen will be accessioned within. Both of these issues have implications for whether or not the new taxa can be made available under the ICZN.

5) There appears to be a minor misunderstanding of the [23] study that established a radiometric age estimate for the deposit (slightly younger than 100 Ma, as opposed to older than 100 Ma), but this does not impact the main findings of the paper, and is also easily addressed.

---

## [Author Response]

*1) Whilst it may be true that this record is the first record of actual eggs and young having been found in an insect showing brood care, the importance is slightly over-emphasised. There are earlier records of what are close to the ortheziids (e.g.,* Burmacoccus*, from the Early Cretaceous, is pretty clearly an ortheziid, and* Cretorthezia*, also from the Early Cretaceous, is very probably an ortheziid, although these taxa are based on adult males). It is highly probable that these two genera also had adult females with an ovisac, which would surely only be used for brood care.*
[15]
*also described an adult female or late nymph that he placed in* Cretorthezia *which looked very similar to extant ortheziids and is therefore very likely to have had an ovisac. Emphasising that* Palaeorthezia *extends crown-group Ortheziidae back only to the mid-Cretaceous therefore seems to overstate their study when there are earlier known probable ortheziid species. This should be explicitly addressed in the manuscript.*

Only two Cretaceous genera have been tentatively attributed to Ortheziidae: *Burmorthezia* and *Cretorthezia.* The genus *Burmorthezia* Vea and Grimaldi, based on second-instar nymphs from mid-Cretaceous Burmese amber, is considered to be an extinct sister group to the crown-group Ortheziidae ([29], page 778: “Cladistically, it is likely that *Burmorthezia* should be assigned to a new family sister to the Ortheziidae…”). The other genus *Cretorthezia* Koteja and Azar was erected based on an adult male from Early Cretaceous Lebanese amber. It has been considered to be a stem group of scale insects ([10], Figure 3) or a group within Ortheziidae ([30], Figure 24). A putative female (‘*Cretorthezia sp*.’, unnamed) was tentatively identified as an ortheziid by [15], and was further discussed by [29]. This female has wax lobes typical of the Ortheziidae, but preserves some strikingly plesiomorphic features (29). Thus, the systematic position of this specimen is still uncertain.

The reviewers also mentioned the genus *Burmacoccus* Koteja, which is based on an adult male from mid-Cretaceous Burmese amber. It belongs to a separate family *Burmacoccidae* (Koteja, 2004), and two recent cladistic analyses also show that it is not closely related to Ortheziidae ([10], Figure 3; [30], Figure 24).

We agreed with the reviewers that the statement that the fossil “extends crown-group Ortheziidae back to the mid-Cretaceous” is “slightly over-emphasised” because *Cretorthezia* from Lebanese amber may be a real ortheziid. We have revised the following to the Discussion:

“Two Cretaceous genera have been tentatively attributed to Ortheziidae: *Burmorthezia* Vea and Grimaldi in mid-Cretaceous Burmese amber is considered […]. Our new fossil supports the view that crown-group Ortheziidae are present in the mid-Cretaceous.”

*2) What is the suggestion about genus/species authorship? Who of the authors of the manuscript will be author of the genus/species? This should be made clear*.

Simon, Szwedo, and Xia are the authors of the genus/species. We have added the genus/species authorship in the main text: “*Wathondara kotejai* Simon, Szwedo, and Xia, 2015”.

*3)*
Figure 4
*is based on studies by at least four other sets of workers and some of the studies were not based on any phylogenetic analyses. The tree is therefore hypothetical and this needs to be stated explicitly. Also,* Palaeorthezia *is not apparently included. Surely it should be. But, if it were, it would not be the earliest ortheziid in this figure! You need to include the names of the genera you are referring to in numbers 5 (Early Cretaceous Lebanese amber), 6 (Mid-Cretaceous Burmese amber) and 7 (Eocene Baltic Amber). This would make it much easier for other workers to follow the arguments in this paper.*

During the review process of our manuscript, a new paper about fossil scale insects has been published (30). It has already presented a comprehensive summary of fossil scale insects, including all taxa and fossil deposits. Thus, we slightly changed our figure to label only several new key taxa that are important for our discussion. We have revised the figure caption of Figure 4.

*4) A significant issue that needs to be addressed is the clear delineation of diagnoses for the new genus and species (even if they are the same), and a clear indication of what collection the specimen will be accessioned within. Both of these issues have implications for whether or not the new taxa can be made available under the ICZN*.

We have added more access information about the specimen to the Materials and methods: “… and will eventually be deposited in the Lingpoge Amber Museum in Shanghai (specimen available for study by contacting B.W. or F.X.).” In this manuscript, we gave only the most important information of this specimen, including a brief diagnosis for the new taxa in this manuscript (the same as Chen et al. *eLife* 2014; 3:e02844), making it available under the ICZN. We will present a detailed description in the following paper that will be published in a specialized journal.

*5) There appears to be a minor misunderstanding of the*
[23]
*study that established a radiometric age estimate for the deposit (slightly younger than 100 Ma, as opposed to older than 100 Ma), but this does not impact the main findings of the paper, and is also easily addressed*.

[23] gave an age for the volcanoclastic matrix of the amber using U-Pb dating of zircons, and they thought that “the age of the volcanic lithic components should be regarded as the age of the amber”. However, some unpublished evidence (e.g., high degree of roundness, bivalve borings on the surface) suggests that the amber was probably reworked before being deposited in the volcanoclastic matrix. Thus, the age of amber may be older than that of the volcanoclastic matrix.

The detailed discussion of the age is beyond the aim of the paper. We agree with the reviewer that “this does not impact the main findings of the paper”. Thus, we have changed the “over 100 million years” to “nearly 100 million years”, and added a brief introduction to Materials and methods: “We tentatively followed the age (98.8 ± 0.6 million years) given by U-Pb dating of zircons from the volcanoclastic matrix of the amber (23). However, some evidence (e.g., high degree of roundness of amber, bivalve borings on the surface of the amber) suggests that the amber was probably reworked before being deposited in the volcanoclastic matrix.”